# Entanglement of photons in their dual wave-particle nature

Adil S. Rab[1], Emanuele Polino[1], Zhong-Xiao Man[2], Nguyen Ba An [3], Yun-Jie Xia[2], Nicolò Spagnolo[1], Rosario Lo Franco [4,5] & Fabio Sciarrino[1]

Wave-particle duality is the most fundamental description of the nature of a quantum object, which behaves like a classical particle or wave depending on the measurement apparatus. On the other hand, entanglement represents nonclassical correlations of composite quantum systems, being also a key resource in quantum information. Despite the very recent observations of wave-particle superposition and entanglement, whether these two fundamental traits of quantum mechanics can emerge simultaneously remains an open issue. Here we introduce and experimentally realize a scheme that deterministically generates entanglement between the wave and particle states of two photons. The elementary tool allowing this achievement is a scalable single-photon setup which can be in principle extended to generate multiphoton wave-particle entanglement. Our study reveals that photons can be entangled in their dual wave-particle behavior and opens the way to potential applications in quantum information protocols exploiting the wave-particle degrees of freedom to encode qubits.

[1] Dipartimento di Fisica, Sapienza Università di Roma, Piazzale Aldo Moro, 5, Roma I-00185, Italy. [2] Shandong Provincial Key Laboratory of Laser Polarization and Information Technology, Department of Physics, Qufu Normal University, Qufu 273165, China. [3] Center for Theoretical Physics, Institute of Physics, Vietnam Academy of Science and Technology (VAST), 18 Hoang Quoc Viet, Cau Giay 10000 Hanoi, Vietnam. [4] Dipartimento di Energia, Ingegneria dell'Informazione e Modelli Matematici, Università di Palermo, Viale delle Scienze, Edificio 9, Palermo 90128, Italy. [5] Dipartimento di Fisica e Chimica, Università di Palermo, via Archirafi 36, Palermo 90123, Italy. Correspondence and requests for materials should be addressed to Z.-X.M. (email: zxman@mail.qfnu.edu.cn) or to R.L.F. (email: rosario.lofranco@unipa.it) or to F.S. (email: fabio.sciarrino@uniroma1.it)

Quantum mechanics is one of the most successful theories in describing atomic-scale systems albeit its properties remain bizarre and counterintuitive from a classical perspective. A paradigmatic example is the wave-particle duality of a single-quantum system, which can behave like both particle and wave to fit the demands of the experiment's configuration[1]. This double nature is well reflected by the superposition principle and evidenced for light by Young-type double-slit experiments[2, 3], where single photons from a given slit can be detected (particle-like behavior) and interference fringes observed (wave-like behavior) on a screen behind the slits. A double-slit experiment can be simulated by sending photons into a Mach–Zehnder interferometer (MZI) via a semitransparent mirror (beam-splitter)[2, 3]. A representative experiment with MZI, also performed with a single atom[4], is the Wheeler's delayed-choice (WDC) experiment[1, 5], where one can choose to observe the particle or wave character of the quantum object after it has entered the interferometer. These experiments rule out the existence of some extra information hidden in the initial state telling the quantum object which character to exhibit before reaching the measurement apparatus. Very recent quantum WDC experiments, using quantum detecting devices and requiring ancilla photons or post-selection, have then shown that wave and particle behaviors of a single photon can coexist simultaneously, with a continuous morphing between them[6–13]. Regarding the latter property, it is worth to mention that the classical concepts of wave and particle need a suitable interpretation in the context of quantum detection. Namely, the wave or particle nature of a photon is operationally defined as the state of the photon, respectively, capable or incapable to produce interference[6]. Along this work, we always retain this operational meaning in terms of two suitably defined quantum states.

When applying the superposition principle to composite systems, another peculiar quantum feature arises, namely the entanglement among degrees of freedom of the constituent particles (e.g., spins, energies, spatial modes, polarizations)[14, 15].

Entanglement gathers fundamental quantum correlations among particle properties, which are at the core of nonlocality[16–20] and exploited as essential ingredient for developing quantum technologies[21–23]. Superposition principle and entanglement have been amply debated within classical-quantum border, particularly whether macroscopically distinguishable states (i.e., distinct quasiclassical wave packets) of a quantum system could be prepared in superposition states[24]. While superpositions of coherent states of a single quantum system (also known as "cat states" from the well-known Schrödinger's epitome) have been observed for optical or microwave fields starting from two decades ago[24–28], the creation of entangled coherent states of two separated subsystems has remained a demanding challenge, settled only very recently by using superconducting microwave cavities and Josephson junction-based artificial atoms[29]. An analogous situation exists in the context of wave-particle duality where, albeit wave-particle superpositions of a photon have been reported[6–12], entangled states of photons correlated in their wave-particle degrees of freedom are still unknown.

In this work we experimentally demonstrate that wave-particle entanglement of two photons is achievable deterministically. We reach this goal by introducing and doubling a scalable all-optical scheme which is capable to generate, in an unconditional manner, controllable single-photon wave-particle superposition states. Parallel use of this basic toolbox then allows the creation of multiphoton wave-particle entangled states.

## Results

**Single-photon toolbox.** The theoretical sketch of the wave-particle scheme for the single photon is displayed in Fig. 1. A photon is initially prepared in a polarization state $|\psi_0\rangle = \cos\alpha|V\rangle + \sin\alpha|H\rangle$, where $|V\rangle$ and $|H\rangle$ are the vertical and horizontal polarization states and $\alpha$ is adjustable by a preparation half-wave plate (not shown in the figure). After crossing the preparation part of the setup of Fig. 1 (see Supplementary

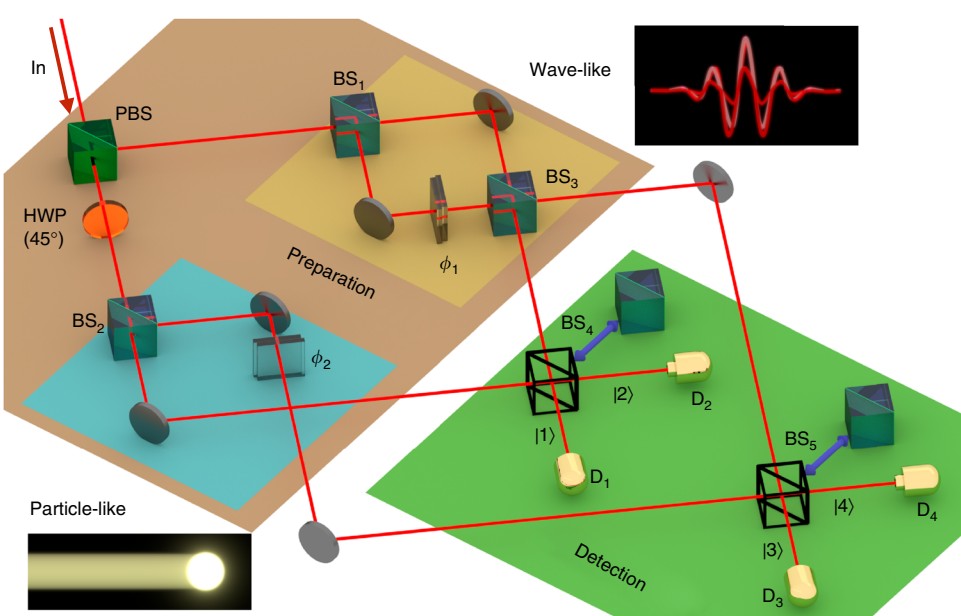

**Fig. 1** Conceptual figure of the wave-particle toolbox. A single photon is coherently separated in two spatial modes by means of a polarizing beam-splitter (PBS) according to its initial polarization state (in). A half-wave plate (HWP) is placed after the PBS to obtain equal polarizations between the two modes. One mode is injected in a complete Mach-Zehnder interferometer (MZI) with phase $\phi_1$, thus exhibiting wave-like behavior. The second mode is injected in a Mach-Zehnder interferometer lacking the second beam-splitter, thus exhibiting particle-like behavior (no dependence on $\phi_2$). The output modes are recombined on two symmetric beam-splitters (BS$_4$, BS$_5$), which can be removed to change the measurement basis. Detectors (D$_1$, D$_2$, D$_3$, D$_4$) are placed on each final path ($|1\rangle$, $|2\rangle$, $|3\rangle$, $|4\rangle$)

Notes 1 and 2 and Supplementary Fig. 1 for details), the photon state is

$$|\psi_f\rangle = \cos\alpha|\text{wave}\rangle + \sin\alpha|\text{particle}\rangle, \quad (1)$$

where the states

$$|\text{wave}\rangle = e^{i\phi_1/2}\left(\cos\frac{\phi_1}{2}|1\rangle - i\sin\frac{\phi_1}{2}|3\rangle\right),$$
$$|\text{particle}\rangle = \frac{1}{\sqrt{2}}\left(|2\rangle + e^{i\phi_2}|4\rangle\right), \quad (2)$$

operationally represent the capacity ($|\text{wave}\rangle$) and incapacity ($|\text{particle}\rangle$) of the photon to produce interference[6, 11]. In fact, for the $|\text{wave}\rangle$ state the probability of detecting the photon in the path $|n\rangle$ ($n = 1, 3$) depends on the phase $\phi_1$: the photon must have traveled along both paths simultaneously (see upper MZI in Fig. 1), revealing its wave behavior. Instead, for the $|\text{particle}\rangle$ state the probability to detect the photon in the path $|n\rangle$ ($n = 2, 4$) is 1/2, regardless of phase $\phi_2$: thus, the photon must have crossed only one of the two paths (see lower MZI of Fig. 1), showing its particle behavior. Notice that the scheme is designed in such a way that $|V\rangle$ ($|H\rangle$) leads to the $|\text{wave}\rangle$ ($|\text{particle}\rangle$) state.

To verify the coherent wave-particle superposition as a function of the parameter $\alpha$, the wave and particle states have to interfere at the detection level. This goal is achieved by exploiting two symmetric beam-splitters where the output paths (modes) are recombined, as illustrated in the detection part of Fig. 1. The probability $P_n = P_n(\alpha, \phi_1, \phi_2)$ of detecting the photon along path $|n\rangle$ ($n = 1, 2, 3, 4$) is now expected to depend on all the involved parameters, namely

$$P_1 = \mathcal{P}_c + \mathcal{I}_c, P_2 = \mathcal{P}_c - \mathcal{I}_c, P_3 = \mathcal{P}_s + \mathcal{I}_s, \ P_4 = \mathcal{P}_s - \mathcal{I}_s, \quad (3)$$

where

$$\mathcal{P}_c = \frac{1}{2}\cos^2\alpha\cos^2\frac{\phi_1}{2} + \frac{1}{4}\sin^2\alpha,$$
$$\mathcal{P}_s = \frac{1}{2}\cos^2\alpha\sin^2\frac{\phi_1}{2} + \frac{1}{4}\sin^2\alpha,$$
$$\mathcal{I}_c = \frac{1}{2\sqrt{2}}\sin 2\alpha\cos^2\frac{\phi_1}{2},$$
$$\mathcal{I}_s = \frac{1}{2\sqrt{2}}\sin 2\alpha\sin\frac{\phi_1}{2}\sin\left(\frac{\phi_1}{2} - \phi_2\right). \quad (4)$$

We remark that the terms $\mathcal{I}_c$, $\mathcal{I}_s$ in the detection probabilities exclusively stem from the interference between the $|\text{wave}\rangle$ and $|\text{particle}\rangle$ components appearing in the generated superposition state $|\psi_f\rangle$ of Eq. (1). This fact is further evidenced by the appearance, in these interference terms, of the factor $\mathcal{C} = \sin 2\alpha$, which is the amount of quantum coherence owned by $|\psi_f\rangle$ in the basis {$|\text{wave}\rangle$, $|\text{particle}\rangle$} theoretically quantified according to the standard $l_1$-norm[30]. On the other hand, the interference terms $\mathcal{I}_c$, $\mathcal{I}_s$ are always identically zero (independently of phase values) when the final state of the photon is: (i) $|\text{wave}\rangle$ ($\alpha = 0$); (ii) $|\text{particle}\rangle$ ($\alpha = \pi/2$); (iii) a classical incoherent mixture $\rho_f = \cos^2\alpha|\text{wave}\rangle\langle\text{wave}| + \sin^2\alpha|\text{particle}\rangle\langle\text{particle}|$ (which can be theoretically produced by the same scheme starting from an initial mixed polarization state of the photon).

The experimental single-photon toolbox, realizing the proposed scheme of Fig. 1, is displayed in Fig. 2 (see Methods for more details). The implemented layout presents the advantage of being interferometrically stable, thus not requiring active phase stabilization between the modes. Figure 3 shows the experimental

results for the measured single-photon probabilities $P_n$. For $\alpha = 0$, the photon is vertically polarized and entirely reflected from the PBS to travel along path 1, then split at $BS_1$ into two paths, both leading to the same $BS_3$ which allows these two paths to interfere with each other before detection. The photon detection probability at each detector $D_n$ ($n = 1, 2, 3, 4$) depends on the phase shift $\phi_1$: $P_1(\alpha = 0) = P_2(\alpha = 0) = \frac{1}{2}\cos^2\frac{\phi_1}{2}$, $P_3(\alpha = 0) = P_4(\alpha = 0) = \frac{1}{2}\sin^2\frac{\phi_1}{2}$, as expected from Eqs. (3) and (4). After many such runs an interference pattern emerges, exhibiting the wave-like nature of the photon. Differently, if initially $\alpha = \pi/2$, the photon is horizontally polarized and, as a whole, transmitted by the PBS to path 2, then split at $BS_2$ into two paths (leading, respectively, to $BS_4$ and $BS_5$) which do not interfere anywhere. Hence, the phase shift $\phi_2$ plays no role on the photon detection probability and each detector has an equal chance to click: $P_1(\alpha = \frac{\pi}{2}) = P_2(\alpha = \frac{\pi}{2}) = P_3(\alpha = \frac{\pi}{2}) = P_4(\alpha = \frac{\pi}{2}) = \frac{1}{4}$, as predicted by Eqs. (3) and (4), showing particle-like behavior without any interference pattern. Interestingly, for $0 < \alpha < \pi/2$, the photon simultaneously behaves like wave and particle. The coherent continuous morphing transition from wave to particle behavior as $\alpha$ varies from 0 to $\pi/2$ is clearly seen from Fig. 4a and contrasted with the morphing observed for a mixed incoherent wave-particle state $\rho_f$ (Fig. 4b). Setting $\phi_2 = 0$, the coherence of the generated state is also directly quantified by measuring the expectation value of an observable $\sigma_x^{1234}$, defined in the four-dimensional basis of the photon paths {$|1\rangle, |2\rangle, |3\rangle, |4\rangle$} of the preparation part of the setup as a Pauli matrix $\sigma_x$ between modes (1, 2) and between modes (3, 4). It is then possible to straightforwardly show that $\langle\sigma_x^{1234}\rangle = \text{Tr}(\sigma_x^{1234}\rho_f) = 0$ for any incoherent state $\rho_f$, while $\sqrt{2}\langle\sigma_x^{1234}\rangle = \sin 2\alpha = \mathcal{C}$ for an arbitrary state of the form $|\psi_f\rangle$ defined in Eq. (1). Insertion of beam-splitters $BS_4$ and $BS_5$ in the detection part of the setup (corresponding to $\beta = 22.5°$ in the output wave-plate of Fig. 2) rotates the initial basis {$|1\rangle, |2\rangle, |3\rangle, |4\rangle$} generating a measurement basis of eigenstates of $\sigma_x^{1234}$, whose expectation value is thus obtained in terms of the detection probabilities as $\langle\sigma_x^{1234}\rangle = P_1 - P_2 + P_3 - P_4$ (see Supplementary Note 2). As shown in Fig. 4c, d, the observed behavior of $\sqrt{2}\langle\sigma_x^{1234}\rangle$ as a function of $\alpha$ confirms the theoretical predictions for both coherent $|\psi_f\rangle$ (Fig. 4c) and mixed (incoherent) $\rho_f$ wave-particle states (the latter being obtained in the experiment by adding a relative time delay in the interferometer paths larger than the photon coherence time to lose quantum interference, Fig. 4d).

**Wave-particle entanglement**. The above single-photon scheme constitutes the basic toolbox which can be extended to create a wave-particle entangled state of two photons, as shown in Fig. 2b. Initially, a two-photon polarization maximally entangled state $|\Psi\rangle_{AB} = \frac{1}{\sqrt{2}}(|VV\rangle + |HH\rangle)$ is prepared (the procedure works in general for arbitrary weights, see Supplementary Note 3). Each photon is then sent to one of two identical wave-particle toolboxes which provide the final state

$$|\Phi\rangle_{AB} = \frac{1}{\sqrt{2}}(|\text{wave}\rangle|\text{wave}'\rangle + |\text{particle}\rangle|\text{particle}'\rangle), \quad (5)$$

where the single-photon states $|\text{wave}\rangle$, $|\text{particle}\rangle$, $|\text{wave}'\rangle$, $|\text{particle}'\rangle$ are defined in Eq. (2), with parameters and paths related to the corresponding wave-particle toolbox. Using the standard concurrence[14] $C$ to quantify the amount of entanglement of this state in the two-photon wave-particle basis, one immediately finds $C = 1$. The generated state $|\Phi\rangle_{AB}$ is thus a wave-particle maximally entangled state (Bell state) of two photons in separated locations.

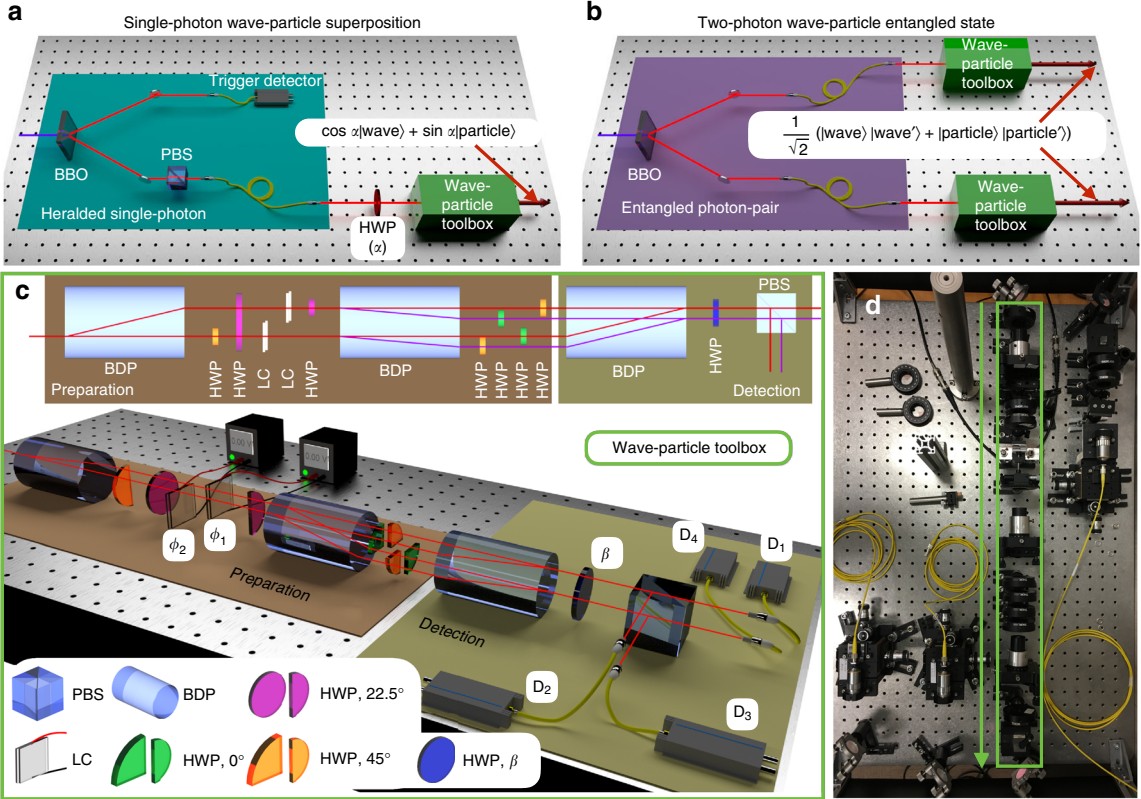

**Fig. 2** Experimental setup for wave-particle states. **a** Overview of the apparatus for the generation of single-photon wave-particle superposition. An heralded single-photon is prepared in an arbitrary linear polarization state through a half-wave plate rotated at an angle $\alpha/2$ and injected into the wave-particle toolbox. **b** Overview of the apparatus for the generation of a two-photon wave-particle entangled state. Each photon of a polarization entangled state is injected into an independent wave-particle toolbox to prepare the output state. **c** Actual implemented wave-particle toolbox, reproducing the action of the scheme shown in Fig. 1. Top subpanel: top view of the scheme, where red and purple lines represent optical paths lying in two vertical planes. Bottom subpanel: 3-d scheme of the apparatus. The interferometer is composed of beam-displacing prisms (BDP), half-wave plates (HWP), and liquid crystal devices (LC), the latter changing the phases $\phi_1$ and $\phi_2$. The output modes are finally separated by means of a polarizing beam-splitter (PBS). The scheme corresponds to the presence of $BS_4$ and $BS_5$ in Fig. 1 for $\beta = 22.5°$, while setting $\beta = 0$ equals to the absence of $BS_4$ and $BS_5$. The same color code for the optical elements (reported in the figure legend) is employed for the top view and for the 3-d view of the apparatus. **d** Picture of the experimental apparatus. The green frame highlights the wave-particle toolbox

The output two-photon state is measured after the two toolboxes. The results are shown in Fig. 5. Coincidences between the four outputs of each toolbox are measured by varying $\phi_1$ and $\phi_1'$. The first set of measurements (Fig. 5a–d) is performed by setting the angles of the output wave-plates (see Fig. 2c) at {$\beta = 0$, $\beta' = 0$}, corresponding to removing both $BS_4$ and $BS_5$ in Fig. 1 (absence of interference between single-photon wave and particle states). In this case, detectors placed at outputs (1, 3) and (1′, 3′) reveal wave-like behavior, while detectors placed at outputs (2, 4) and (2′, 4′) evidence a particle-like one. As expected, the two-photon probabilities $P_{nn'}$ for the particle detectors remain unchanged while varying $\phi_1$ and $\phi_1'$, whereas the $P_{nn'}$ for the wave detectors show interference fringes. Moreover, no contribution of crossed wave-particle coincidences $P_{nn'}$ is obtained, due to the form of the entangled state. The second set of measurements (Fig. 5e–h) is performed by setting the angles of the output wave-plates at {$\beta = 22.5°$, $\beta' = 22.5°$}, corresponding to the presence of $BS_4$ and $BS_5$ in Fig. 1 (the presence of interference between single-photon wave and particle states). We now observe nonzero contributions across all the probabilities depending on the specific settings of phases $\phi_1$ and $\phi_1'$. The presence of entanglement in the wave-particle behavior is also assessed by measuring the quantity $\mathcal{E} = P_{22'} - P_{21'}$ as a function of $\phi_1$, with

fixed $\phi_1' = \phi_2 = \phi_2' = 0$. According to the general expressions of the coincidence probabilities (see Supplementary Note 3), $\mathcal{E}$ is proportional to the concurrence $C$ and identically zero (independently of phase values) if and only if the wave-particle two-photon state is separable (e.g., $|\text{wave}\rangle \otimes |\text{wave}'\rangle$) or a maximal mixture of two-photon wave and particle states. For $|\Phi\rangle_{AB}$ of Eq. (5) the theoretical prediction is $\mathcal{E} = (1/4)\cos^2(\phi_1/2)$, which is confirmed by the results reported in Fig. 5i, j (within the reduction due to visibility). A further test of the generated wave-particle entanglement is finally performed by the direct measure of the expectation values $\langle \mathcal{W} \rangle = \text{Tr}(\mathcal{W}\rho)$ of a suitable entanglement witness[31], defined in the $(4 \times 4)$-dimensional space of the two-photon paths as

$$\mathcal{W} = \mathbb{1} - 2\left[\sigma_x^{1234} \otimes \left(\sigma_x^{1234}\right)'\right] - \left[\sigma_z^{1234} \otimes \left(\sigma_z^{1234}\right)'\right], \quad (6)$$

where $\mathbb{1}$ is the identity matrix, $\sigma_x^{1234}$ has been defined previously, and $\sigma_z^{1234}$ corresponds to applying a $\sigma_z$ Pauli matrix between modes (1, 2) and between modes (3, 4). The measurement basis of $\sigma_z^{1234}$ is that of the initial paths {$|1\rangle, |2\rangle, |3\rangle, |4\rangle$} exiting the preparation part of the single-photon toolbox. It is possible to show that $\text{Tr}(\mathcal{W}\rho_s) \geq 0$ for any two-photon separable state $\rho_s$ of wave-particle states, so that whenever $\text{Tr}(\mathcal{W}\rho_e) < 0$ the state $\rho_e$ is

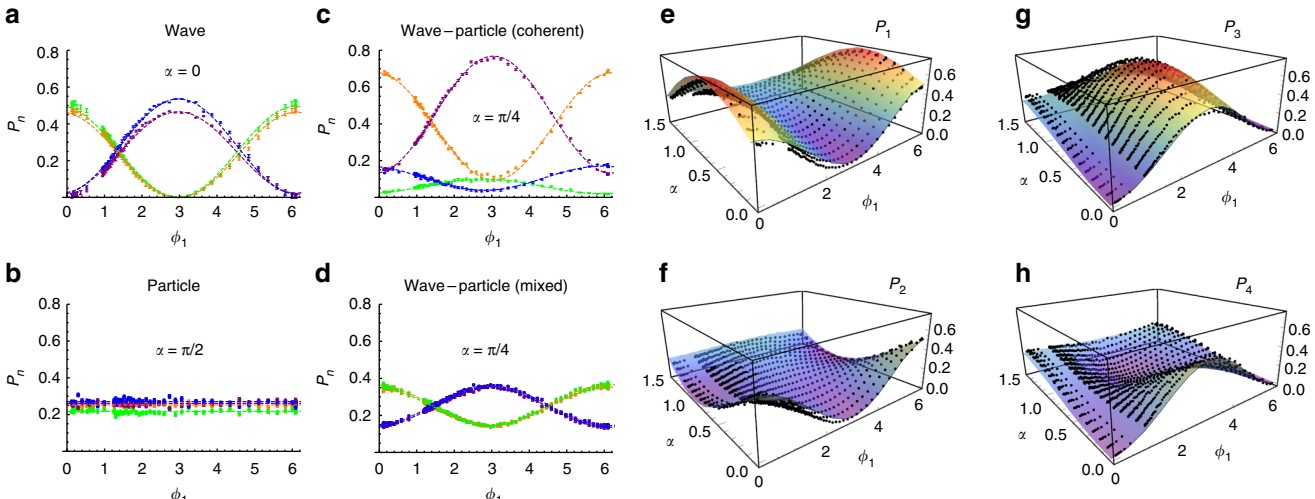

**Fig. 3** Generation of wave-particle superposition with a single-photon state. **a–d** Measurements of the output probabilities $P_n$ as a function of the phase $\phi_1$, for different values of $\alpha$. **a** Wave behavior ($\alpha = 0$). **b** Particle behavior ($\alpha = \pi/2$). **c** Coherent wave-particle superposition ($\alpha = \pi/4$). **d** Incoherent mixture of wave and particle behaviors ($\alpha = \pi/4$). Points: experimental data. Dashed curves: best-fit of the experimental data. Color legend: orange ($P_1$), green ($P_2$), purple ($P_3$), blue ($P_4$). **e–h** 3-d plots output probabilities $P_n$ as a function of the phase $\phi_1$ and of the angle $\alpha$. Points: experimental data. Surfaces: theoretical expectations. In all plots, error bars are standard deviation due to the Poissonian statistics of single-photon counting

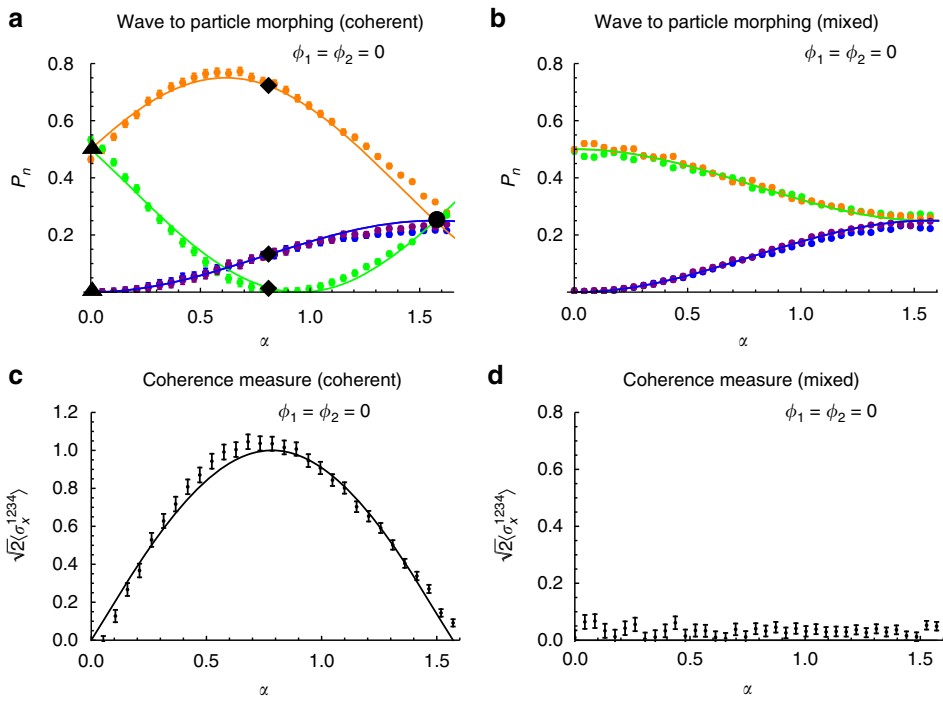

**Fig. 4** Evidence of the generation of wave-particle superpositions. **a** Probabilities $P_n$ as a function of $\alpha$ in the coherent case and **b** for an incoherent mixture. Color legend: orange ($P_1$), green ($P_2$), purple ($P_3$), blue ($P_4$). In **a** black triangles highlighted the position for wave behavior ($\alpha = 0$), black circle for particle behavior ($\alpha = \pi/2$) and black diamonds highlight the position for coherent wave-particle superposition behavior ($\alpha = \pi/4$). **c** Coherence measure $\sqrt{2}\langle\sigma_x^{1234}\rangle$ as a function of $\alpha$ in the coherent case and **d** for an incoherent mixture (the latter showing no dependence on $\alpha$). Points: experimental data. Solid curves: theoretical expectations. Error bars are standard deviations due to the Poissonian statistics of single-photon counting

entangled in the photons wave-particle behavior (see Supplementary Note 3). The expectation values of $\mathcal{W}$ measured in the experiment in terms of the 16 coincidence probabilities $P_{nn'}$, for the various phases considered in Fig. 5, are: $\langle\mathcal{W}\rangle = -0.699 \pm 0.041$ ($\phi_1 = \phi_1' = 0$); $\langle\mathcal{W}\rangle = -0.846 \pm 0.045$ ($\phi_1 = \phi_1' = \pi$); $\langle\mathcal{W}\rangle = -0.851 \pm 0.041$ ($\phi_1 = \pi$, $\phi_1' = 0$); $\langle\mathcal{W}\rangle = -0.731 \pm 0.042$ ($\phi_1 = 0$, $\phi_1' = \pi$). These observations altogether

prove the existence of quantum correlations between wave and particle states of two photons in the entangled state $|\Phi\rangle_{AB}$.

## Discussion

In summary, we have introduced and realized an all-optical scheme to deterministically generate single-photon wave-particle superposition states. This setup has enabled the observation of the

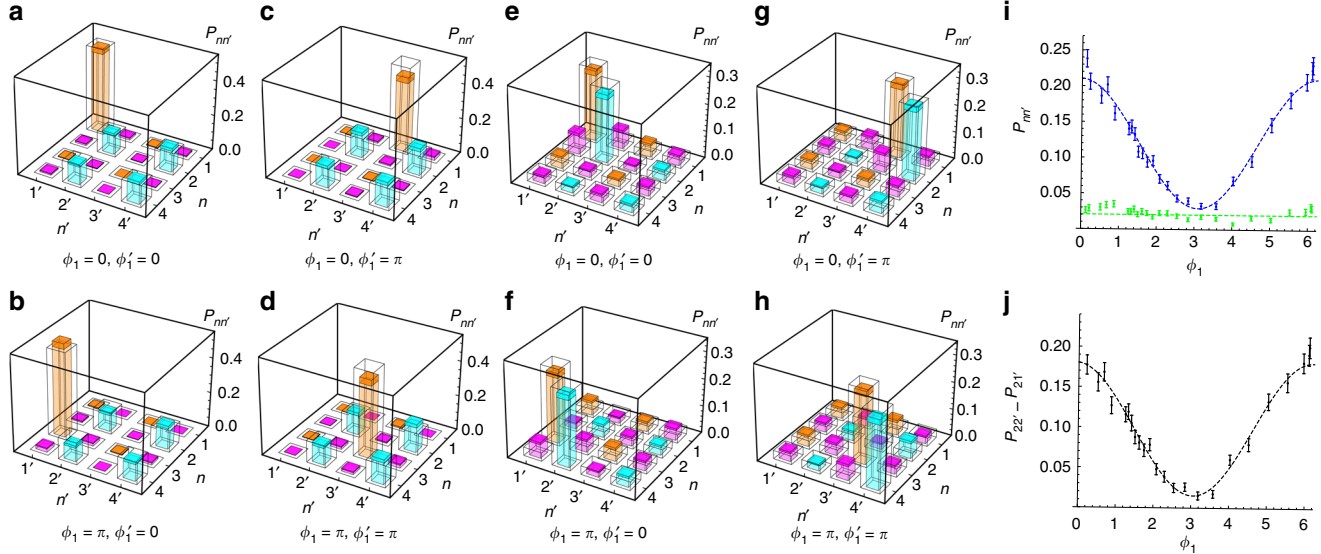

**Fig. 5** Generation of wave-particle entangled superposition with a two-photon state. Measurements of the output coincidence probabilities $P_{nn'}$ to detect one photon in output mode $n$ of the first toolbox and one in the output mode $n'$ of the second toolbox, with different phases $\phi_1$ and $\phi'_1$ ($\phi_2 = \phi'_2 = 0$). **a–d**, $P_{nn'}$ measured with {$\beta = 0$, $\beta' = 0$}, corresponding to the absence of BS$_4$ and BS$_5$ in Fig. 1. **a** $\phi_1 = 0$ and $\phi'_1 = 0$. **b** $\phi_1 = \pi$ and $\phi'_1 = 0$. **c** $\phi_1 = 0$ and $\phi'_1 = \pi$. **d** $\phi_1 = \pi$ and $\phi'_1 = \pi$. **e–h** $P_{nn'}$ measured with {$\beta = 22.5°$, $\beta' = 22.5°$}, corresponding to the presence of BS$_4$ and BS$_5$ in Fig. 1. **e** $\phi_1 = 0$ and $\phi'_1 = 0$. **f** $\phi_1 = \pi$ and $\phi'_1 = 0$. **g** $\phi_1 = 0$ and $\phi'_1 = \pi$. **h** $\phi_1 = \pi$ and $\phi'_1 = \pi$. White bars: theoretical predictions. Colored bars: experimental data. Orange bars: $P_{nn'}$ contributions for detectors D$_n$ and D$_{n'}$ linked to wave-like behavior for both photons (in the absence of BS$_4$ and BS$_5$). Cyan bars: $P_{nn'}$ contributions for detectors D$_n$ and D$_{n'}$ linked to particle-like behavior for both photons (in the absence of BS$_4$ and BS$_5$). Magenta bars: $P_{nn'}$ contributions for detectors D$_n$ and D$_{n'}$ linked to wave-like behavior for one photon and particle-like behavior for the other one (in absence of BS$_4$ and BS$_5$). Darker regions in colored bars correspond to 1 $\sigma$ error interval, due to the Poissonian statistics of two-photon coincidences. **i**, **j**, Quantitative verification of wave-particle entanglement. **i**, $P_{22'}$ (blue) and $P_{21'}$ (green) and **j**, $\mathcal{E} = P_{22'} - P_{21'}$, as a function of $\phi_1$ for $\phi'_1 = 0$ and {$\beta = 22.5°$, $\beta' = 22.5°$}. Error bars are standard deviations due to the Poissonian statistics of two-photon coincidences. Dashed curves: best-fit of the experimental data

simultaneous coexistence of particle and wave character of the photon maintaining all its devices fixed, being the control only on the preparation of the input photon. Specifically, different initial polarization states of the photon, then transformed into which-way (path) states, reveal the wave-to-particle morphing economizing the employed resources compared with previous experiments with delayed choice[6–12]. The advantageous aspects of the single-photon scheme have then supplied the key for its straightforward doubling, by which we have observed that two photons can be cast in a wave-particle entangled state provided that suitable initial entangled polarization states are injected into the apparatus. We remark that powerful features of the scheme are flexibility and scalability. Indeed, a parallel assembly of $N$ single-photon wave-particle toolboxes allows the generation of $N$-photon wave-particle entangled states. For instance, the GHZ-like state $|\Phi_N\rangle = \frac{1}{\sqrt{2}}(|\text{wave}_1, \text{wave}_2, \ldots, \text{wave}_N\rangle + |\text{particle}_1, \text{particle}_2, \ldots, \text{particle}_N\rangle)$ is produced when the GHZ polarization entangled state $|\Psi_N\rangle = \frac{1}{\sqrt{2}}(|V_1 V_2 \ldots V_N\rangle + |H_1 H_2 \ldots H_N\rangle)$ is used as input state.

From the viewpoint of the foundations of quantum mechanics, our research brings the complementarity principle for wave-particle duality to a further level. Indeed, it merges this basic trait of quantum mechanics with another peculiar quantum feature such as the entanglement. In fact, besides confirming that a photon can live in a superposition of wave and particle behaviors when observed by quantum detection[11], we prove that the manifestation of its dual behavior can intrinsically depend on the dual character of another photon, according to correlations ruled by quantum entanglement. Specifically, the coherent wave-particle behavior of a photon is quantum correlated to the measurement outcome of an apparatus, sensitive to the wave-

particle behavior of another photon, placed in a region separated from it. Our work shows that this type of entanglement is possible for composite quantum systems. We finally highlight that the possibility to create and control wave-particle entanglement may also play a role in quantum information scenarios. In particular, it opens the way to design protocols which exploit quantum resources contained in systems of qubits encoded in wave and particle operational states.

## Methods

**Experimental wave-particle toolbox.** The implementation of the wave-particle toolbox exploits both polarization and path degrees of freedom of the photons. A crucial parameter is to obtain an implemented toolbox presenting high interferometric stability. This is achieved in the experiment by exploiting the scheme of Fig. 2, which presents an intrinsic interferometric stability due to the adoption of calcite crystals as beam-displacing prisms (see Supplementary Note 1). More specifically, all optical paths of the overall interferometer are transmitted by the same beam-displacing prisms and propagate in parallel directions, and are thus affected by the same phase fluctuations. Relative phases $\phi_1$ and $\phi_2$ (Fig. 2) within the interferometer are controlled by two liquid crystal devices, which introduce a tunable relative phase between polarization state $|H\rangle$ and $|V\rangle$ depending on the applied voltage. The parameter $\alpha$ of Eq. (1) is set by an input half-wave plate, while the output half-wave plate at the detection stage rotates the measurement basis depending on its angle $\beta$ ($\beta = 0°$ corresponds to the absence of BS$_4$ and BS$_5$, while $\beta = 22.5°$ corresponds to the presence of BS$_4$ and BS$_5$). Both half-wave plates are controlled by a motorized stage. Hence, all the variable optical elements in the setup can be controlled via software.

**Acquisition system.** The output photons are detected by avalanche photodiode detectors, which are connected to an id800 Time to Digital Converter from ID Quantique that is employed to record the output single counts and two-photon coincidences. The photon source is a parametric down conversion source generating pairs of entangled photons. In the single particle experiment, one of the generated photon is directly detected and acts as a trigger, while the other photon is injected in the wave-particle toolbox. Two-photon coincidences are recorded between the output detectors of the toolbox and the trigger photon. In the two-

particle experiment, the two photons of the entangled pair are separately sent to two independent wave-particle toolboxes. Two-photon coincidences are then recorded between the output detectors of each toolbox. A dedicated LabVIEW routine allows simultaneous control of the optical elements and of the detection apparatus to obtain a fully automatized measurement process.

**Data availability**. The data sets generated during and/or analyzed during the current study are available from the corresponding author on reasonable request.

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

## Acknowledgements

This work was supported by the ERC-Starting Grant 3D-QUEST (3D-Quantum Integrated Optical Simulation; grant agreement no. 307783, http://www.3dquest.eu) and by the Marie Curie Initial Training Network PICQUE (Photonic Integrated Compound Quantum Encoding, grant agreement no. 608062, funding Program: FP7-PEOPLE-2013-ITN, http://www.picque.eu). In this work Z.-X.M. and Y.-J.X. are supported by the National Natural Science Foundation of China under Grant Nos. 11574178 and 61675115, Shandong Provincial Natural Science Foundation, China under Grant No. ZR2016JL005, while N.B.A. is funded by the Vietnam National Foundation for Science and Technology Development (NAFOSTED) under project no. 103.01-2017.08.

## Author contributions

Z.-X.M., N.B.A., Y.-J.X. and R.L.F. devised the theoretical proposal. A.S.R., E.P., N.S. and F.S. designed and performed the experiment. R.L.F. and F.S. coordinated the project. All the authors discussed the results and contributed to the preparation of the manuscript.

## Additional information

**Competing interests:** The authors declare no competing financial interests.

