## [Peer Review File · Nature Communications]

Reviewers' comments:

Reviewer #1 (Remarks to the Author):

The authors report experiments in which they investigate the simultaneous wave-like and particle-like nature of photons. The main experimental apparatus for these investigations, the 'wave-particle toolbox' is an interferometer that maps an incoming photon to either a closed or open interferometer, accord to the state of its polarisation. A photon in the closed interferometer is said to behave like a wave, while a photon in the open interferometer is said to behave like a particle. These two arms are then closed as part of an overall interferometer. Tuning phase shifters inside these interferometers produces fringes in photon detection counts that are consistent with photons in superpositions across the interferometer.

Two wave-particle toolboxes are used with a source of polarisation entangled photon pairs. Pairs of photons are then said to be entangled in wave/particle degrees of freedom. data for correlated photon detection is consistent with theoretical predictions for entangled photons under the described operations. The authors conclude by highlighting that they have demonstrated wave-particle duality at a distance.

The experimental techniques, conclusions, analysis and write up are sound. Determining the impact of this work is a case of assessing how interesting or thought provoking it is to entangle particles in their complementarity. I do not find anything particularly surprising in the work, though I can imagine those working in foundations of quantum mechanics might find it interesting. It may therefore be suited to publication in Nature Communications.

Referee report on NCOMMS-17-05795-T – Entanglement of photons in their dual wave-particle nature

The manuscript presents an extension of the quantum delayed-choice Gedankenexperiment proposed in [11] and experimentally realized in [12-18]. The authors propose, and experimentally realise, a wave-particle entangled state of two photons produced by SPDC. The article is well written and the results are timely in our quest of understanding the puzzling aspects of quantum mechanics – there is a recent resurgence of activity in quantum foundations. The experiment is well-done, with an ingenious design which makes the interferometers stable (using beam displacing prisms).

I have a few comments.

1. It is not clear why we need the beamsplitters BS_4 , BS_5 . Without them, the output states in eq.(2) are:

$$|wave'\rangle = \frac{e^{i\frac{\phi_1}{2}}}{\sqrt{2}} \left(\cos \frac{\phi_1}{2} |1\rangle - i \sin \frac{\phi_1}{2} |3\rangle \right) \quad (1)$$

$$|particle'\rangle = \frac{1}{2} (|2\rangle + e^{i\phi_2} |4\rangle) \quad (2)$$

similar to the ones defined in ref.[11]; this makes easier to see that $\langle wave'|particle'\rangle = 0$. In Supplementary Information it is claimed that BS_4 , BS_5 are required in order to demonstrate the wave-particle morphing. However, morphing can be shown from the above equations by varying α , e.g., as experimentally shown in ref.[12]-[16].

2. There is a discrepancy in the description of mixed states:

pg.3: "a classical incoherent mixture ρ_f [...] (resulting from an initial mixed polarization state of the photon)".

pg.4: "[...] mixed ρ_f wave-particle states (the latter being obtained by adding a relative time delay in the interferometer paths larger than the photon coherence time to lose quantum interference)".

Which of these two methods are used in the experiment?

3. pg.4: "According to equations (3) and (4), $W_C = 2I_c$ is zero if and only if there is no wave-particle coherence."

However, $W_C = 0$ if: (i) there is no wave-particle coherence, $C = \sin 2\alpha = 0$, or (ii) $\phi_1 = \pi$. Thus "if and only if" is not correct. Is there an intuition of why $W_C = 0$ for $\phi_1 = \pi$?

It will be useful to have an understanding of the coherence witness W_C . The authors just define it as $W_C = |P_1 - P_2|$ without details. According to ref.[30]: "A coherence witness [...] is a Hermitian operator W such that $\text{Tr}[W\sigma] \geq 0$ is true for all incoherent states $\sigma \in \mathcal{I}$ ". It will be useful to include in the

Supplementary material a derivation of $W_C = |P_1 - P_2|$ (or at least a reference to this).

4. Supplementary Information, pg.4.: "We point out that the third terms of all the coincidence probabilities of equation (S17) are zero if and only if there is no quantum entanglement between the wave and particle degrees of freedom of the two photons".

Again, "if and only if" is an overstatement. The third terms are also zero if, e.g., $\phi_1 = \phi'_1 = \pi$ and $\phi_2 + \phi'_2 = 3\pi/2$. What is so special about these values – is there a physical intuition behind them?

Similarly, "if and only if" after eq. (S18), the definition of the entanglement witness W_E .

5. Personally I find Supplementary Fig.1 better at conveying the experiment than Fig.2c from the main text. The heavy 3D graphics is more difficult to read than its 2D counterpart in Supp.Info. Although it's fashionable nowadays to have 3D models, these are sometimes less informative than the 2D ones. As such, they fail their main purpose: to better convey information to the reader.

To conclude, I would recommend publication, provided the above questions/comments are answered satisfactory.

Reply to the Reviewers' reports of
"Entanglement of photons in their dual wave-particle nature"
(NCOMMS-17-05795-T) by Adil S. Rab et al.

General reply

We thank the Reviewers for their comments which highlight both the fundamental interest of our results and the effectiveness of the experimental design. In this revised version of the manuscript, we have made changes in order to comply with the Reviewers' comments. In the following, we list our detailed reply to the Reviewers and the changes made to the manuscript. We believe this revised version of the manuscript may be suitable for publication in Nature Communications.

Reviewer's comments

Reviewer #1 (Remarks to Author):

1. **Reviewer # 1:** *The authors report experiments in which they investigate the simultaneous wave-like and particle-like nature of photons. The main experimental apparatus for these investigations, the 'wave-particle toolbox' is an interferometer that maps an incoming photon to either a closed or open interferometer, accord to the state of its polarisation. A photon in the closed interferometer is said to behave like a wave, while a photon in the open interferometer is said to behave like a particle. These two arms are then closed as part of an overall interferometer. Tuning phase shifters inside these interferometers produces fringes in photon detection counts that are consistent with photons in superpositions across the interferometer.*

Two wave-particle toolboxes are used with a source of polarisation entangled photon pairs. Pairs of photons are then said to be entangled in wave/particle degrees of freedom. data for correlated photon detection is consistent with theoretical predictions for entangled photons under the described operations. The authors conclude by highlighting that they have demonstrated wave-particle duality at a distance.

The experimental techniques, conclusions, analysis and write up are sound. Determining the impact of this work is a case of assessing how interesting or thought provoking it is to entangle particles in their complementarity. I do not find anything particularly surprising in the work, though I can imagine those working in foundations of quantum mechanics might find it interesting. It may therefore be suited to publication in Nature Communications.

Our reply. We thank the Reviewer for noticing that our results are sound and they can be interesting within the foundations of quantum mechanics. Indeed, our experiment merges two fundamental traits of quantum theory. On one side, wave-particle duality represents the fundamental description of a quantum object whose behaviour depends on the measurement apparatus. On the other side, entanglement is one of the most peculiar properties arising as nonclassical correlations between distant objects. In our experiment, we show the emergence of entanglement in the dual wave-particle nature of two quantum particles. Our research brings the complementarity particle to a new level, showing the possibility to determine the wave-particle behaviour of a system by acting at distance on a separate object in its wave-particle characteristics by using a suitable measurement apparatus. In order to further point out the role of our work within the foundations of quantum mechanics, we have extended the final paragraph of the manuscript. See, in particular, page 5, left column:

"From the viewpoint of the foundations of quantum mechanics, our research brings the complementarity principle for wave-particle duality to a further level. Indeed, it merges this basic trait of quantum mechanics with another peculiar quantum feature such as the entanglement. In fact, besides confirming that a photon can live in a superposition of wave and particle behaviors when observed by quantum detection [11], we prove that the manifestation of its dual nature can intrinsically depend on the dual character of another photon, according to correlations ruled by quantum entanglement. In this case the wave-particle behavior of a photon is determined by a measurement apparatus, sensitive to the wave-particle nature of another photon, placed in a region spatially separated from it. This phenomenon can be named "wave-particle duality action at a distance" and constitutes a new fundamental property of composite quantum systems. "

Reviewer #2 (Remarks to Author):

Reviewer # 2: *The manuscript presents an extension of the quantum delayed-choice Gedankenexperiment proposed in [11] and experimentally realized in [12-18]. The authors propose, and experimentally realise, a wave-particle entangled state of two photons produced by SPDC. The article is well written and the results are timely in our quest of understanding the puzzling aspects of quantum mechanics - there is a recent resurgence of activity in quantum foundations. The experiment is well-done, with an ingenious design which makes the interferometers stable (using beam displacing prisms).*

Our reply. We thank the Reviewer for pointing out that our work is of significant interest and original. In the following, we give our replies to all the Reviewer's points.

1. **Reviewer #2:** *It is not clear why we need the beamsplitters BS_4 , BS_5 . Without them, the output states in eq. (2) are:*

$$|wave'\rangle = \frac{e^{i\frac{\phi_1}{2}}}{\sqrt{2}} \left(\cos \frac{\phi_1}{2} |1\rangle - i \sin \frac{\phi_1}{2} |3\rangle \right) \quad (1)$$

$$|particle'\rangle = \frac{1}{2} (|2\rangle + e^{i\phi_2} |4\rangle) \quad (2)$$

similar to the ones defined in ref.[11]; this makes easier to see that $\langle wave'|particle'\rangle = 0$. In Supplementary Information it is claimed that BS_4 , BS_5 are required in order to demonstrate the wave-particle morphing. However, morphing can be shown from the above equations by varying α , e.g., as experimentally shown in ref.[12]-[16].

Our reply. The Reviewer is right in noting that, without beamsplitters BS_4 and BS_5 , the state would be a coherent superposition of states $|wave'\rangle$ and $|particle'\rangle$, named in Eq. (S4) of Supplementary Information as wave-like and particle-like states (see also the states inside parentheses in the second line after Eq. (S4)) and the wave-particle morphing can still be realized by varying α . However, from the experimental viewpoint, the photon properties are examined statistically via detection probabilities. In fact, the detections are performed in the orthogonal basis $\{|wave'\rangle, |particle'\rangle\}$ in absence of BS_4 and BS_5 . Namely, as we write in the Supplementary Information after Eq. (S4), the photon wave-like behaviour is confirmed by detections at D_1 and D_3 with a total probability of $\cos^2 \alpha$ while the particle-like behaviour is confirmed at D_2 and D_4 with a total probability of $\sin^2 \alpha$. However, the same probabilities would be also obtained if the photon were in a classically mixed state $\rho_f = \cos^2 \alpha |wave'\rangle\langle wave'| + \sin^2 \alpha |particle'\rangle\langle particle'|$ and there would be, at this stage, no way to distinguish the coherent and mixed states from one another. Therefore, to experimentally verify the photon wave-particle coherent pure state, we add BS_4 and BS_5 to observe interferences of the wave and particle components directly from the detection probabilities (see Fig. 3 (a) and Fig. 3(c) of the main text). The possibility to remove and insert the beam-splitters BS_4 and BS_5 also plays a crucial role in the wave-particle entanglement, since it permits the change of measurement basis and thus the observation of quantum wave-particle correlations in the generated entangled state (see Fig. 4 of the main text).

In order to make this aspect clearer, we have modified the sentences starting at the last line of page 1 of the Supplementary Information, namely

"However, these counting probabilities do not allow us to observe a wave-to-particle morphing since there is no interference between the wave-like and particle-like states at the detection level. It is like measuring the state $|\psi_3\rangle$ only along the orthogonal basis corresponding to wave-like or particle-like behavior.

In order to observe a wave-to-particle morphing as a function of the parameter α , the wave-like and particle-like behaviors have to interfere at the detection level. This is achieved by [...]"

with the following sentences:

"However, these counting probabilities would not allow us to distinguish the wave-particle morphing corresponding to the coherent superposition of wave-like and particle-like behaviors of equation (S4) from that corresponding to their classical mixture with probabilities $\cos^2 \alpha$ and $\sin^2 \alpha$, respectively. It is like measuring the state $|\psi_3\rangle$ only along the orthogonal basis corresponding to wave-like or particle-like behavior.

In order to observe a coherent wave-to-particle morphing as a function of the parameter α , the wave-like and particle-like behaviors have to interfere at the detection level. This is achieved by [...]"

We have also modified a sentence in the main text, page 4, left column, second paragraph, line 12 from the bottom:

"The coherent continuous morphing transition from wave to particle behavior as α varies from 0 to $\pi/2$ is clearly seen from Fig. 3c and contrasted with the morphing observed for a mixed wave-particle state ρ_f ".

2. **Reviewer # 2:** *There is a discrepancy in the description of mixed states: pg.3: "a classical incoherent mixture ρ_f [...] (resulting from an initial mixed polarization state of the pho-*

ton)”.

pg.4: “[...] mixed ρ_f wave-particle states (the latter being obtained by adding a relative time delay in the interferometer paths larger than the photon coherence time to lose quantum interference)”.

Which of these two methods are used in the experiment?

Our reply. We have modified the manuscript and the Supplementary Information to remove ambiguities on this aspect. The first description (that is, “resulting from an initial mixed polarization state of the photon”) is just a theoretical one which can be useful to arrive at the final mixed state ρ_f by following the theoretical scheme. The second description is the method effectively utilized in the experiment to get the mixed state. In order to avoid this ambiguity, we have changed the sentences, respectively, as:

- Page 4 of the main text, left column, line 2,
”a classical incoherent mixture $\rho_f = \cos^2 \alpha |\text{wave}\rangle \langle \text{wave}| + \sin^2 \alpha |\text{particle}\rangle \langle \text{particle}|$ (which can be theoretically produced by the same scheme starting from an initial mixed polarization state of the photon)”.
- Page 4 of the main text, left column, second paragraph, line 5 from the bottom,
”[...] and mixed ρ_f wave-particle states (the latter being obtained in the experiment by adding a relative time delay in the interferometer paths larger than the photon coherence time to lose quantum interference).”
- At the end of the first section of the Supplementary Information (Supplementary Note 1, page 2), we have added the sentence: ”In the experiment, such a state is effectively achieved by adding a relative time delay in the interferometer paths larger than the photon coherence time to lose quantum interference. Comparisons of the wave-particle morphing and of the coherence witness between the case of coherent wave-particle superposition and the case of mixed wave-particle state are reported in the manuscript (see Fig. 3).”

3. **Reviewer # 2:** pg.4: ”According to equations (3) and (4), $\mathcal{W}_C = 2\mathcal{I}_C$ is zero if and only if there is no wave-particle coherence”.

However, $\mathcal{W}_C = 0$ if: (i) there is no wave-particle coherence, $C = \sin 2\alpha = 0$, or (ii) $\phi_1 = \pi$. Thus, ”if and only if” is not correct. Is there an intuition of why $\mathcal{W}_C = 0$ for $\phi_1 = \pi$?

It will be useful to have an understanding of the coherence witness \mathcal{W}_C . The authors just define it as $\mathcal{W}_C = |P_1 - P_2|$ without details. According to ref.[30]: ”A coherence witness [...] is a Hermitian operator W such that $\text{Tr}[W\sigma] \geq 0$ is true for all incoherent states $\sigma \in \mathcal{I}$ ”. It will be useful to include in the Supplementary material a derivation of $\mathcal{W}_C = |P_1 - P_2|$ (or at least a reference to this).

Our reply. We agree with the Reviewer that \mathcal{W}_C is zero for $\phi_1 = \pi$. Due to the interference effect at BS₃, when $\phi_1 = \pi$ the photon will only exit from path 3, while path 1 has no photon, so no interference will occur with path 2 at BS₄. Since the interference term \mathcal{I}_c of Eq. (4) is induced by the interference between path 1 and path 2, which does not occur for $\phi_1 = \pi$, one has $\mathcal{I}_c = 0$ and thus $\mathcal{W}_C = 2|\mathcal{I}_c| = 0$. Indeed, the writing ”if and only if” was not precise in this case, since we instead meant that the witness \mathcal{W}_C is identically zero, that is independently of phase values, if and only if there is no wave-particle coherence. With this specification, the claim ”if and only if” is justified. The definition of this wave-particle coherence witness just stems from combining the experimental detection probabilities we have at our hands in order that the final quantity results proportional (up to some function of the phases) to the standard coherence quantifier $C = \sin 2\alpha$ in the orthogonal wave-particle basis. For instance, we could also define a coherence witness as the sum of \mathcal{I}_c and \mathcal{I}_s of Eq. (4) for $\phi_2 = 0$ (whose value is irrelevant for the wave-particle interference being involved in the particle-like part of the setup), obtaining a quantity proportional to $C = \sin 2\alpha$ and independent of ϕ_1 . However, for our aim, it was sufficient to testify that the wave-particle state is coherent and we have done this for a specific value of $\phi_1 = 0$, which makes \mathcal{W}_C proportional to the coherence and contrasting it with the behavior for a mixed wave-particle state (see Fig. 3c).

In order to make this point clearer and avoid confusion, we have modified the sentence in the main text, line 8 after Eq. (4) as:

”On the other hand, the interference terms \mathcal{I}_c and \mathcal{I}_s are always identically zero (independently of phase values) when [...]”.

We have also changed the sentence at page 4, left column, second paragraph, line 8 from the bottom:

”According to equations (3) and (4), $\mathcal{W}_C = 2|\mathcal{I}_c|$ is identically zero, independently of the phase values, if and only if there is no wave-particle coherence.”

Moreover, following the Reviewer’s suggestion, we have added in the Supplementary Information a brief description of how the coherence witness is constructed, according to the lines explained above. See, in particular, Supplementary Note 1, page 2, line 9 after Eq. (S7):

”In the main text, we then define a wave-particle coherence witness $\mathcal{W}_C = |P_1 - P_2| = 2|\mathcal{I}_c|$ in terms of

some of the detection probabilities. This definition of \mathcal{W}_C just stems from the experimental demand of combining the observed detection probabilities in order that the final quantity results proportional (up to some function of the phases) to the standard coherence quantifier $\mathcal{C} = \sin 2\alpha$. In fact, a quantity so constructed is experimentally employable as a witness of wave-particle quantum coherence being identically zero, that is independently of the values of the phases, if and only if there is no coherence between the $|\text{wave}\rangle$ and $|\text{particle}\rangle$ components. We have therefore chosen a straightforward way to obtain this quantity on the basis of the theoretical expressions of the detection probabilities (see equations (3) and (4) of the main text). ”

4. **Reviewer # 2:** *Supplementary Information, pg.4.: ”We point out that the third terms of all the coincidence probabilities of equation (S17) are zero if and only if there is no quantum entanglement between the wave and particle degrees of freedom of the two photons.” Again, ”if and only if” is an overstatement. The third terms are also zero if, e.g., $\phi_1 = \phi'_1 = \pi$ and $\phi_2 + \phi'_2 = 3\pi/2$. What is so special about these values - is there a physical intuition behind them? Similarly, ”if and only if” after eq. (S18), the definition of the entanglement witness \mathcal{W}_E .*

Our reply. This issue is analogous to the previous one for the single-photon experiment and again comes from the lack of specifying that the third terms of all the coincidence probabilities of equation (S17) are identically zero (independently of the phase values) if and only if there is no quantum entanglement between the wave and particle degrees of freedom of the two photons. We have now made this at page 4 of the Supplementary Information (before Eq. (S18)). Regarding the entanglement witness, which is defined in the same spirit of the coherence witness described above, we correctly write ”identically zero” both in the main text and the Supplementary Information. However, we have modified in the main text, page 4, right column, line 11 from the bottom of the paragraph before the Discussion section, the sentence: ”According to the general expressions of the coincidence probabilities (see Supplementary Information), \mathcal{W}_E is identically zero (independently of phase values) if and only if [...]”

We can also give the reason why the entanglement witness is zero at the values of phases noticed by the Reviewer. When $\phi_1 = \phi'_1 = \pi$, no photons will exit from paths 1 and 1' so that no interferences appear at BS_4 and $\text{BS}_{4'}$. Therefore, all the third terms of the coincidence probabilities in the first six equations of (S17) associated with the detectors D_1, D'_1, D_2 and D'_2 vanish. When both $\phi_1 = \phi'_1 = \pi$ and $\phi_2 + \phi'_2 = 3\pi/2$ at the same time, no interferences at BS_5 and BS'_5 appear either, so the third terms of the last two equations of (S17) also vanish. By the way, we note that, besides the above-mentioned special values of phases, there are some other ones, such as $\phi_1 = 0$ and/or $\phi'_1 = 0$, that can induce the same vanishing effect in some of the third terms of Eq. (S17). Nevertheless, the witness is identically zero for all the values of the phases if and only if there is no quantum entanglement between the wave and particle degrees of freedom.

5. **Reviewer # 2:** *Personally I find Supplementary Fig.1 better at conveying the experiment than Fig.2c from the main text. The heavy 3D graphics is more difficult to read than its 2D counterpart in Supp.Info. Although it's fashionable nowadays to have 3D models, these are sometimes less informative than the 2D ones. As such, they fail their main purpose: to better convey information to the reader.*

Our reply. We have modified Figure 2 in the main text to take into account the comment by Reviewer # 2. In the new version, panel c of Figure 2 now shows both a 2-d top view of the scheme (as shown in Supplementary Fig. 1 b) and the 3-d view shown in the previous version of the Figure. This will help the reader to get all the necessary information on the experimental apparatus.

Reviewers' comments:

Reviewer #3 (Remarks to the Author):

Rab et al.'s manuscript "Entanglement of photons in their dual wave-particle nature" exposes an interesting experiment showing an entangled state which is a coherent superposition of having two photons which can show interference when a phase is changed (the $|wave\rangle$ state), or two photons that cannot (the $|particle\rangle$ state). Overall, I agree with reviewer 1 that while there is nothing particularly surprising about this work, it might be of some interest to those working on quantum foundations.

In this revised version the authors have responded to the issues raised by reviewer 2. Here I will provide my opinion on the revisions to address these points (1-5) before adding a few comments of my own (6-7). To summarize, while most of reviewer 2's concerns have been addressed, there remain two points (3b and 4b) regarding the construction of the coherence witness and the entanglement witness which I think need to be addressed before I can recommend publication.

1. The need for BS4 and BS5

The authors specify in their response that this is required to distinguish a coherent superposition of $|particle\rangle$ and $|wave\rangle$ states from an incoherent mixture, and have updated the supplementary information accordingly. I find this explanation and change to the manuscript satisfactorily addresses the issue.

2. Discrepancy in the description of mixed states.

The manuscript has been updated to make clear that the experiment uses a relative time delay to produce mixed states, whereas the mixed polarisation of the input photon is a theoretical example. I believe this clarifies the ambiguity raised by reviewer 2.

3. Issues with coherence witness

a. Conditions for being zero

The mention of "independently of phase value" has been added, thus clarifying the if and only if statements.

b. Derivation of the coherence witness W_c .

I find that the answer this point raises more questions than it answers. As reviewer 2 pertinently pointed out, reference [30] defines a coherence witness as "a Hermitian operator W such that $\text{Tr}[W\sigma] \geq 0$ is true for all incoherent states $\sigma \in I$ ". I do not see how W_c fits this definition.

In their response, the authors state that W_c is a valid witness because, according to equations (3) and (4), it results in a value proportional to $\sin(\alpha)$, which should be "the amount of quantum coherence owned by [the state] in the basis wave/particle".

However, knowledge of the state should not be used to determine if a given Hermitian operator is a coherence witness. Therefore, using equations 3 and 4, which come from a-priori knowledge of the state, cannot be used to justify the use of W_c as a coherence witness.

Furthermore, the statement that $C = \sin(\alpha)$ represents "the amount of quantum coherence (...) in the basis wave/particle", is justified using reference [30]. That paper discusses several different measures which can be used to quantify coherence. Which one is used here to get $C = \sin(\alpha)$? Or, if the measure used is not from that paper, does it respect conditions (C1-C4) from that paper?

4. Issues with the entanglement witness

a. Conditions for being zero.

The mention of "independently of phase value" has been added, clarifying the "if and only if" statements.

b. Construction of the entanglement witness (not explicitly stated by reviewer 2, but the authors address it in their response).

Once again, I find the article's explanation lacking on this point. It is not clear at all to me that W_e is an entanglement witness, at least according to the standard definition: "An observable W is called an entanglement witness if $\text{Tr}(W \rho_s) \geq 0$ for all separable states ρ_s , and $\text{Tr}(W \rho_e) < 0$ for at least one entangled state ρ_e " [O. Gühne and G. Tóth, Physics Reports 474, 1 (2009)].

The problem is that appropriateness of W_e as a witness is justified using prior knowledge of their state, which is contrary to the spirit of entanglement witnesses. Equation (S18) can only be obtained with prior knowledge that the states have the form of (S16). Furthermore, is it really possible to confirm bipartite entanglement using only two outcomes of a single measurement (P22' and P21')? This seems analogous to concluding that two qubits are entangled using only an X-basis measurement – wouldn't more measurements (or assumptions about the state) be required? It might be possible to construct a true entanglement witness using additional information from the other measurement performed without the beam-splitters, but this would have to be justified.

5. Fig 2c.

I wholeheartedly agree with reviewer 2 – I do not find that the three dimensional helps convey relevant information to the reader. The addition of the 2D projection in the main text does represent an improvement.

6. Nuances in discussion

In the discussion, it might be a good idea to rework the following sentence: "In this case, the wave-particle behavior of a photon is determined by a measurement apparatus (...) placed in a region spatially separated from it." As currently written, it might give readers the impression that non-local signaling is possible.

I also feel like the claim that "wave-particle duality action at a distance" is a "fundamental property of composite quantum" is a bit too strong. This experiment uses a very specific setup to produce entanglement with states called $|\text{particle}\rangle$ and $|\text{wave}\rangle$. This is hardly a fundamental property. Perhaps simply highlighting this paper's demonstration that this type of entanglement is possible would be more reasonable.

7. Wave/particle language

I find the way the words particle and wave are used in this article could be misleading. As pointed out in reference [6]: "Classical concepts like particle or wave (as in "wave-particle duality") do not translate perfectly into the quantum language. For example, although we observe interference (a definite wavelike behavior), the pattern is produced click-by-click, in a discrete, particlelike manner." This imperfect translation leads them to define an "operational definition of "wave" or "particle" to stand for "ability" or "inability" to produce interference". In other words, wave and particle can have two possible meanings in this context. A classical meaning, which refers to how object behave when they goes through slits, whether they are localised, how they are detected, etc.; as stated in reference [6], this is what is meant when one talks about "wave-particle duality". The second meaning, which is introduced as an operational definition, is compatible with quantum mechanics and is what makes it possible to talk about concepts such as $|\text{wave}\rangle/|\text{particle}\rangle$ superposition or $|\text{wave}\rangle/|\text{particle}\rangle$ entanglement.

The distinction is important because a photon in the $|\text{particle}\rangle$ state is of course not a "particle", it

is still a quantum-mechanical object which possesses "wave-particle duality" – for example, if slits were placed in its path we could still observe an interference pattern afterwards. Similarly, photons in the $|\text{wave}\rangle$ state are still detected as single photons, a particle-like behavior. So $|\text{particle}\rangle \neq$ "particle" and $|\text{wave}\rangle \neq$ "wave".

The issue I would like to raise is that wave and particle are used throughout the paper to refer to both these meanings interchangeably. For example, the abstract mentions wave-particle duality - which, as stated in [6], refers to the fact that quantum systems have behaviours associated with the classical concepts of waves and particles) - then goes on to discuss $|\text{wave}\rangle/|\text{particle}\rangle$ superposition and $|\text{wave}\rangle/|\text{particle}\rangle$ entanglement, which are only well defined based on the operational definition of these states. I do not think it is reasonable to expect that the typical Nature Communications reader will notice this subtlety.

This becomes even more problematic for sentences in the introduction and discussion where the two meanings seem to mix, taking results which have to do with operationally defined $|\text{wave}\rangle$ and $|\text{particle}\rangle$ quantum states and applying them to the "wave" and "particle" nature of photons. Take for example, "entangled in the dual wave-particle nature". As discussed above, photons still have a "dual wave-particle nature" when they are in the $|\text{particle}\rangle$ or $|\text{wave}\rangle$ states. So how does it make sense to talk about entanglement of these properties? Of course, the entanglement is meant to refer to the $|\text{particle}\rangle$ and $|\text{wave}\rangle$ states so in that context it makes perfect sense, but this operational meaning is not what is typically implied by "dual wave-particle nature" of photons.

To address the issue and ensure that readers are not misled, I strongly recommend that the operational definition of the $|\text{particle}\rangle$ and $|\text{wave}\rangle$ states be given as early as possible in the paper rather than on page 3 in the results section, and that the authors find a way to remove any ambiguity as to the meanings of these words throughout the paper, avoiding in particular the attribution of the properties of the $|\text{particle}\rangle$ and $|\text{wave}\rangle$ states to the dual particle-wave nature of photons.

Reply to the Reviewers' reports of
"Entanglement of photons in their dual wave-particle nature"
(NCOMMS-17-05795-A) by Adil S. Rab et al.

General reply

We wish to thank the Reviewer #3 for Her/His careful analysis of our previous version in replying to the first Reviewers' reports. In this new revised version of the manuscript, we have made modifications to comply with the Reviewer's criticisms and suggestions. In particular, the discussion related to the coherence and entanglement witness has led us to identify suitable directly measurable observables of these properties independent of the knowledge of the state. This further analysis thus makes our results even stronger. In the following, we list our detailed reply to the Reviewer and the changes made to the manuscript. We further include in the submission a pdf files with all the highlighted changes performed on the manuscript. We believe this improved version of the manuscript may be now suitable for publication in Nature Communications.

Reviewer's comments

Reviewer #3 (Remarks to Author):

Reviewer # 3:

Rab et al.'s manuscript "Entanglement of photons in their dual wave-particle nature" exposes an interesting experiment showing an entangled state which is a coherent superposition of having two photons which can show interference when a phase is changed (the $|\text{wave}\rangle$ state), or two photons that cannot (the $|\text{particle}\rangle$ state). Overall, I agree with reviewer 1 that while there is nothing particularly surprising about this work, it might be of some interest to those working on quantum foundations.

In this revised version the authors have responded to the issues raised by reviewer 2. Here I will provide my opinion on the revisions to address these points (1-5) before adding a few comments of my own (6-7). To summarize, while most of reviewer 2's concerns have been addressed, there remain two points (3b and 4b) regarding the construction of the coherence witness and the entanglement witness which I think need to be addressed before I can recommend publication.

Our reply. We thank the Reviewer for the careful reading and for the useful comments. We now respond point-by-point to the remaining queries.

Reviewer #3:

1) *The need for BS_4 and BS_5 The authors specify in their response that this is required to distinguish a coherent superposition of $|\text{particle}\rangle$ and $|\text{wave}\rangle$ states from an incoherent mixture, and have updated the supplementary information accordingly. I find this explanation and change to the manuscript satisfactorily addresses the issue.*

2) *Discrepancy in the description of mixed states. The manuscript has been updated to make clear that the experiment uses a relative time delay to produce mixed states, whereas the mixed polarisation of the input photon is a theoretical example. I believe this clarifies the ambiguity raised by reviewer 2.*

3) *Issues with coherence witness.*

3a) Conditions for being zero. *The mention of "independently of phase value" has been added, thus clarifying the if and only if statements.*

Our reply. We acknowledge that the Reviewer is satisfied with our previous answer to these points.

Reviewer # 3:

3b) Derivation of the coherence witness \mathcal{W}_c . *I find that the answer this point raises more questions than it answers. As reviewer 2 pertinently pointed out, reference [30] defines a coherence witness as "a Hermitian operator \mathcal{W} such that $\text{Tr}[\mathcal{W}] \geq 0$ is true for all incoherent states $\sigma \in I$ ". I do not see how \mathcal{W}_c fits this definition. In their response, the authors state that \mathcal{W}_c is a valid witness because, according to equations (3) and (4), it results in a value proportional to $\sin(\alpha)$, which should be "the amount of quantum coherence owned by [the state] in the basis wave/particle". However, knowledge of the state should not be used to determine if a given Hermitian operator is a coherence witness. Therefore, using equations 3 and 4, which come from a-priori knowledge of the state, cannot be used to justify the use of \mathcal{W}_c as a coherence witness. Furthermore,*

the statement that $C = \sin(\alpha)$ represents "the amount of quantum coherence (...) in the basis wave/particle", is justified using reference [30]. That paper discusses several different measures which can be used to quantify coherence. Which one is used here to get $C = \sin(\alpha)$? Or, if the measure used is not from that paper, does it respect conditions (C1-C4) from that paper?

Our reply. We thank the Reviewer for pointing out this issue. We reckon that the use of the term "witness" was confusing, due to the usual meaning given in the literature to a coherence witness. From a theoretical perspective, we refer to the bona-fide coherence measure obtained by the standard l_1 -norm, defined in Eq. (35) of Ref. [30]. We have now clarified that the generated wave-particle coherent state is the one before beam-splitters BS_4 and BS_5 (see Eqs. (1) and (2) of the present manuscript), being the latter exploited for the detection of the state (as shown in Fig. 1). The l_1 -norm coherence quantifier just give the theoretical result $C = \sin(2\alpha)$ for the coherence of this state. Regarding the experimental measure of the wave-particle coherence, we have now defined an observable whose expectation value gives the coherence of the wave-particle state and is directly measurable by the photon counting probabilities. This experimentally friendly observable does not require the knowledge of the state and provides a direct quantitative measure of the wave-particle coherence of the generated state.

We have thus made modifications in both manuscript and Supplementary information, as follows:

- Figure 3c. Coherence witness has been changed in "Coherence measure". In the caption, the previously named coherence witness has now become: "coherence measure $\sqrt{2}\sigma_x^{1234}$ ".
- Page 3, we have changed the explicit expressions of the wave and particle operational states of Eq. (2).
- Page 3, right column, line 5. We have modified the paragraph introducing the detection probabilities as: "To verify the coherent wave-particle superposition as a function of the parameter α , the wave and particle states have to interfere at the detection level. This goal is achieved by exploiting two symmetric beam-splitters where the output paths (modes) are recombined, as illustrated in the detection part of Fig.1). The probability $P_n = P_n(\alpha, \phi_1, \phi_2)$ of detecting the photon along path $|n\rangle$ ($n = 1, 2, 3, 4$) is now expected to depend on all the involved parameters, namely..."
- Page 4, left column, lines 7-8 from the top. We have added the sentence: "[...] theoretically quantified according to the standard l_1 -norm [30]"
- Page 4, last 15 lines of left column and first 6 lines of right column, before section "Wave-particle entanglement". We have changed the final paragraph dedicated to the measure of coherence as follows: "Setting $\phi_2 = 0$, the coherence of the generated state is also directly quantified by measuring the expectation value of an observable σ_x^{1234} , defined in the 4-dimensional basis of the photon paths $\{|1\rangle, |2\rangle, |3\rangle, |4\rangle\}$ of the preparation part of the setup as a Pauli matrix σ_x between modes (1,2) and between modes (3,4). It is then possible to straightforwardly show that $\langle \sigma_x^{1234} \rangle = \text{Tr}(\sigma_x^{1234} \rho_f) = 0$ for any incoherent state ρ_f , while $\sqrt{2}\langle \sigma_x^{1234} \rangle = \sin 2\alpha = C$ for an arbitrary state of the form $|\psi_f\rangle$ defined in Eq. (1). Insertion of beam-splitters BS_4 and BS_5 in the detection part of the setup (corresponding to $\beta = 22.5^\circ$ in the output wave-plate of Fig.2) rotates the initial basis $\{|1\rangle, |2\rangle, |3\rangle, |4\rangle\}$ generating a measurement basis of eigenstates of σ_x^{1234} , whose expectation value is thus obtained in terms of the detection probabilities as $\langle \sigma_x^{1234} \rangle = P_1 - P_2 + P_3 - P_4$ (see Supplementary Information). As shown in Fig.3c, the observed behaviour of $\sqrt{2}\langle \sigma_x^{1234} \rangle$ as a function of α confirms the theoretical predictions for both coherent $|\psi_f\rangle$ and mixed (incoherent) ρ_f wave-particle state"
- Supplementary Information, pages 1-3, starting from Eq. (S4). We have changed and extended the first Section of the Supplementary Information, mainly to define and describe the σ_x^{1234} observable for the direct measurement of the wave-particle coherent state (see, in particular, from Eq. (S6) to Eq. (S10) and the subsequent arguments until the first paragraph of page 3).

Reviewer # 3:

4) *Issues with the entanglement witness*

4a) **Conditions for being zero.** *The mention of "independently of phase value" has been added, clarifying the "if and only if" statements.*

Our reply. We acknowledge that the Reviewer is satisfied with our previous answer to this point.

Reviewer # 3:

4b) **Construction of the entanglement witness** *(not explicitly stated by reviewer 2, but the authors address it in their response). Once again, I find the article's explanation lacking on this point. It is not clear*

at all to me that W_e is an entanglement witness, at least according to the standard definition: "An observable W is called an entanglement witness if $\text{Tr}(W\rho_s) \geq 0$ for all separable states ρ_s , and $\text{Tr}(W\rho_e) < 0$ for at least one entangled state ρ_e " [O. Gühne and G. Tóth, *Physics Reports* 474, 1 (2009)]. The problem is that appropriateness of W_e as a witness is justified using prior knowledge of their state, which is contrary to the spirit of entanglement witnesses. Equation (S18) can only be obtained with prior knowledge that the states have the form of (S16). Furthermore, is it really possible to confirm bipartite entanglement using only two outcomes of a single measurement ($P_{22'}$ and $P_{21'}$)? This seems analogous to concluding that two qubits are entangled using only an X -basis measurement - wouldn't more measurements (or assumptions about the state) be required? It might be possible to construct a true entanglement witness using additional information from the other measurement performed without the beam-splitters, but this would have to be justified.

Our reply. We again thank the Reviewer for pointing out this issue. The spirit of our reply is the same of that given above for the measure of coherence. Our use of the term "entanglement witness" was indeed confusing with respect to the usual meaning this term is given in quantum information, as the Reviewer opportunely highlights. For the theoretical entanglement quantifier, we refer to the standard concurrence of a known two-qubit state and we then construct, in terms of the detected probabilities, a quantity which is proportional to the concurrence. Of course, this approach requires the knowledge of the generated state. We stress that the presence of a wave-particle entanglement is already assessed by the observations of all the coincidence probabilities, satisfying the correlations expected for an entangled state in the two-photon wave-particle basis, as we firstly explain in the manuscript (page 4, right column, paragraph starting from line 9 after Eq. (6)). The experimental measure of a quantity proportional to the concurrence serves as a quantitative demonstration of the generated entangled state in wave particle basis. However, stimulated by the Reviewer's comments, we have made an effort to verify the presence of the wave-particle entangled state by a faithful entanglement witness, directly measurable, without resorting to knowledge of the state. In fact, in this new version of the manuscript we have: (i) found an entanglement witness in terms of combinations of local observables σ_x^{1234} and σ_z^{1234} (see new Eq. (7) of the manuscript), proving that it satisfies the requirements of a reliable witness for a state in wave particle basis and citing to this purpose O. Gühne and G. Tóth, *Physics Reports* 474, 1 (2009) (see new Ref. [31]); (ii) performed a direct measurement of the expectation value of this witness by inserting and removing the additional beam-splitters (BS_4 , BS_5 , BS_4 , BS_5) of each detection part of the setup. This measurement results in a suitable combination of the detected coincidence probabilities. Our observations give negative values of the witness (see page 5, right column, paragraph before "Discussion" section) which thus guarantee the presence of wave-particle entanglement.

In order to make everything clear regarding these new aspects, we have made changes in both manuscript and Supplementary Information, as follows:

- Page 4, line 3 after Eq. (5). We have added the sentence: "Using the standard concurrence C [14] to quantify the amount of entanglement of this state in the two-photon wave-particle basis, one immediately finds $C = 1$."
- Page 4, right column, line 10 from the bottom until the end of section at page 5 right column, before section "Discussion". We have changed and extended the final paragraph of section dedicated to the wave-particle entanglement as: "The presence of entanglement in the wave-particle behaviour is also assessed by measuring the quantity $\mathcal{E} = P_{22'} - P_{21'}$ as a function of ϕ_1 , with fixed $\phi'_1 = \phi_2 = \phi'_2 = 0$. According to the general expressions of the coincidence probabilities (see Supplementary Information), \mathcal{E} is proportional to the concurrence C and identically zero (independently of phase values) if and only if the wave-particle two-photon state is separable (e.g., $|\text{wave}\rangle \otimes |\text{wave}'\rangle$) or a maximal mixture of two-photon wave and particle states). For $|\Phi\rangle_{AB}$ of equation (5) the theoretical prediction is $\mathcal{E} = (1/4) \cos^2(\phi_1/2)$, which is confirmed by the results reported in Fig.4i-j (within the reduction due to visibility). A further test of the generated wave-particle entanglement is finally performed by the direct measure of the expectation values $\langle W \rangle = \text{Tr}(W\rho)$ of a suitable entanglement witness [31], defined in the (4×4) -dimensional space of the two-photon paths as [...]. These observations altogether prove the existence of quantum correlations between wave and particle states of two photons in the entangled state $|\Phi\rangle_{AB}$."
- Page 5, caption of Fig. 4. We have slightly changed the sentences regarding panels (i-j) of Fig. 4 as follows: Quantitative verification of wave-particle entanglement. **i**, $P_{22'}$ (blue) and $P_{21'}$ (green) and **j**, $\mathcal{E} = P_{22'} - P_{21'}$, as a function of [...].
- Supplementary Information, pages 3 and 4. We have redefined Eqs. (S15), (S16), (S20), (S21) in order to clearly individuate the generated wave-particle entangled state and the state transformed for the detection along the rotated wave-particle basis.
- Supplementary Information, pages 4-6, starting at line 7 after Eqs. (S23) until page 6 before Section "Supplementary Note 3". We have changed and extended this paragraph dedicated to the measure of

entanglement, introducing the entanglement witness and providing the proof that it satisfies the requirements for a proper witness (see, in particular, the arguments after Eq. (S25)). The new part begins and ends as follows: "We point out that the third terms of all the coincidence probabilities of equation (S17) are identically zero (that is, independently of the values of the phases) if and only if there is no quantum entanglement between the wave and particle degrees of freedom of the two photons. [...] These observations ultimately confirm the effective generation of a wave-particle entanglement".

Reviewer # 3:

5) *Fig 2c. I wholeheartedly agree with reviewer 2 I do not find that the three dimensional helps convey relevant information to the reader. The addition of the 2D projection in the main text does represent an improvement.*

Our reply. We acknowledge that the Reviewer is satisfied with our previous answer to this point.

Reviewer # 3:

6) *Nuances in discussion. In the discussion, it might be a good idea to rework the following sentence: "In this case, the wave-particle behavior of a photon is determined by a measurement apparatus (...) placed in a region spatially separated from it." As currently written, it might give readers the impression that non-local signaling is possible. I also feel like the claim that "wave-particle duality action at a distance" is a "fundamental property of composite quantum" is a bit too strong. This experiment uses a very specific setup to produce entanglement with states called $|\text{particle}\rangle$ and $|\text{wave}\rangle$. This is hardly a fundamental property. Perhaps simply highlighting this paper's demonstration that this type of entanglement is possible would be more reasonable.*

Our reply. Following the Reviewer's suggestion, we have rephrased the sentences in object in order to make the claims sounder. In particular, the sentences are now as follows: "Specifically, the coherent wave-particle behaviour of a photon is quantum correlated to the measurement outcome of an apparatus, sensitive to the wave-particle nature of another photon, placed in a region separated from it. Our work shows that this type of entanglement is possible for composite quantum systems."

Reviewer # 3:

7) *Wave/particle language. I find the way the words particle and wave are used in this article could be misleading. As pointed out in reference [6]: "Classical concepts like particle or wave (as in "wave-particle duality") do not translate perfectly into the quantum language. For example, although we observe interference (a definite wavelike behavior), the pattern is produced click-by-click, in a discrete, particlelike manner" This imperfect translation leads them to define an "operational definition of "wave" or "particle" to stand for "ability" or "inability" to produce interference". In other words, wave and particle can have two possible meanings in this context. A classical meaning, which refers to how object behave when they goes through slits, whether they are localised, how they are detected, etc.; as stated in reference [6], this is what is meant when one talks about "wave-particle duality". The second meaning, which is introduced as an operational definition, is compatible with quantum mechanics and is what makes it possible to talk about concepts such as $|\text{wave}\rangle/|\text{particle}\rangle$ superposition or $|\text{wave}\rangle/|\text{particle}\rangle$ entanglement.*

The distinction is important because a photon in the $|\text{particle}\rangle$ state is of course not a "particle", it is still a quantum-mechanical object which possesses "wave-particle duality" for example, if slits were placed in its path we could still observe an interference pattern afterwards. Similarly, photons in in the $|\text{wave}\rangle$ state are still detected as single photons, a particle-like behavior. So $|\text{particle}\rangle \neq$ "particle" and $|\text{wave}\rangle \neq$ "wave".

The issue I would like to raise is that wave and particle are used throughout the paper to refer to both these meanings interchangeably. For example, the abstract mentions wave-particle duality - which, as stated in [6], refers to the fact that quantum systems have behaviours associated with the classical concepts of waves and particles) - then goes on to discuss $|\text{wave}\rangle/|\text{particle}\rangle$ superposition and $|\text{wave}\rangle/|\text{particle}\rangle$ entanglement, which are only well defined based on the operational definition of these states. I do not think it is reasonable to expect that the typical Nature Communications reader will notice this subtlety.

This becomes even more problematic for sentences in the introduction and discussion where the two meanings seem to mix, taking results which have to do with operationally defined $|\text{wave}\rangle$ and $|\text{particle}\rangle$ quantum states and applying them to the "wave" and "particle" nature of photons. Take for example, "entangled in the dual wave-particle nature". As discussed above, photons still have a "dual wave-particle nature" when they are in the $|\text{particle}\rangle$ or $|\text{wave}\rangle$ states. So how does it make sense to talk about entanglement of these properties? Of course, the entanglement is meant to refer to the $|\text{particle}\rangle$ and $|\text{wave}\rangle$ states so in that context it makes perfect sense, but this operational meaning is not what is typically implied by "dual wave-particle nature" of photons. To address the issue and ensure that readers are not mislead, I strongly recommend that the operational def-

inition of the $|\text{particle}\rangle$ and $|\text{wave}\rangle$ states be given as early as possible in the paper rather than on page 3 in the results section, and that the authors find a way to remove any ambiguity as to the meanings of these words throughout the paper, avoiding in particular the attribution of the properties of the $|\text{particle}\rangle$ and $|\text{wave}\rangle$ states to the dual particle-wave nature of photons.

Our reply. We thank the Reviewer for pointing out this point. On the light of the Reviewer's comments, we fully agree that a clarification of the meaning of wave or particle nature of a photon in the context of quantum detection is better given as early as possible in the manuscript. We have then modified the manuscript to improve the clarity of this aspect.

More specifically, we have added a new paragraph in the introduction, page 1, left column, just after Refs. [6-13]: "Regarding the latter property, it is worth to mention that the classical concepts of wave and particle need a suitable interpretation in the context of quantum detection. Namely, the wave or particle nature of a photon is operationally defined as the state of the photon, respectively, capable or incapable to produce interference [6]. Along this work, we always retain this operational meaning in terms of two suitably defined quantum states."

We believe that such a paragraph is useful to the reader to understand the very meaning of the wave and particle behaviour of a photon within the quantum detection context and avoid any ambiguity of interpretation along the paper.

Moreover, the explicit definitions of the $|\text{wave}\rangle$ and $|\text{particle}\rangle$ states in equations (2) further supply the reader with an explicit definition of the operational dual (wave-particle) nature of the photon in this context.

Finally, we modified where necessary the text (see for instance point 6 of the reviewer) to avoid misleading interpretations.

REVIEWERS' COMMENTS:

Reviewer #3 (Remarks to the Author):

In this revised version of the manuscript, it is my opinion that Rab et al. have satisfactorily addressed the issue raised by Reviewer 2 and myself. I am particularly pleased that the authors have been able to find and measure a suitable entanglement witness to confirm entanglement of the state to further strengthen their conclusions. I recommend publication in Nature Communications.